# Compact and Low-Profile UWB Antenna Based on Graphene-Assembled Films for Wearable Applications

**DOI:** 10.3390/s20092552

**Published:** 2020-04-30

**Authors:** Ran Fang, Rongguo Song, Xin Zhao, Zhe Wang, Wei Qian, Daping He

**Affiliations:** 1Hubei Engineering Research Center of RF-Microwave Technology and Application, Wuhan University of Technology, Wuhan 430070, China; fangran@whut.edu.cn (R.F.); rongguo_song@whut.edu.cn (R.S.); xzhao@whut.edu.cn (X.Z.); wangzhe0614@whut.edu.cn (Z.W.); qianwei@whut.edu.cn (W.Q.); 2State Key Laboratory of Advanced Technology for Materials Synthesis and Processing, Wuhan University of Technology, Wuhan 430070, China

**Keywords:** graphene-assembled film, UWB, CPW, wearable antenna

## Abstract

In this article, a graphene-assembled film (GAF)-based compact and low-profile ultra-wide bandwidth (UWB) antenna is presented and tested for wearable applications. The highly conductive GAFs (~10^6^ S/m) together with the flexible ceramic substrate ensure the flexibility and robustness of the antenna, which are two main challenges in designing wearable antennas. Two H-shaped slots are introduced on a coplanar-waveguide (CPW) feeding structure to adjust the current distribution and thus improve the antenna bandwidth. The compact GAF antenna with dimensions of 32 × 52 × 0.28 mm^3^ provides an impedance bandwidth of 60% (4.3–8.0 GHz) in simulation. The UWB characteristics are further confirmed by on-body measurements and show a bending insensitive bandwidth of ~67% (4.1–8.0 GHz), with the maximum gain at 7.45 GHz being 3.9 dBi and 4.1 dBi in its flat state and bent state, respectively. Our results suggest that the proposed antenna functions properly in close proximity to a human body and can sustain repetitive bending, which make it well suited for applications in wearable devices.

## 1. Introduction

Wearable wireless technologies have been attracting increasing attention for solving the conformal problem between wearable antennas and clothes [1,2]. With the rapid growth of the wearable equipment market, various flexible wearable devices have already been widely used in the wireless communication, medical monitoring, and military fields [3,4,5]. This has stimulated extensive research on wearable antennas that are flexible, compact, integratable with clothing, and robust [6,7,8]. Ultra-wide bandwidth (UWB) communication covers a frequency bandwidth of 7.5 GHz (from 3.1 to 10.6 GHz). Due to its extremely low range of effective isotropic power emission density, the battery life of wearable and implantable health monitoring devices can be longer. In addition, UWB communication also provides good adaptability for multipath effects and potential applications [9,10,11]. UWB technology is important to improve the performance of antenna in wearable wireless applications due to its merits of high data transmission rate, broad bandwidth, and low-power short-range communication, which is usually realized by fractal and multi-layer structural design [12,13,14]. Based on different structural designs, UWB technology has been used in many wearable antenna applications, but the miniaturization of antennas and the improvement of radiation efficiency are still worthy of research [15,16]. Sun et al. proposed a cloth with 100% cotton used as the substrate in the design of a UWB textile antenna. To emulate the on-body condition, they studied the antenna performance when it is in bent and crumpled conditions, and also when the feed line and ground plane are misaligned because of human body movement. However, the radiation efficiency of the antenna was poor, at less than 50% [15]. Simorangkir et al. presented a flexible UWB antenna for wearable applications. The antenna had a physical size of 80 mm × 67 mm, which failed to meet the design requirements for miniaturization of wearable antennas [17]. Coplanar-waveguide (CPW) feeding is also a common method to improve the bandwidth and realize miniaturization [18,19,20,21,22,23]. Great progress has been achieved for traditional antennas to solve the problems relating to antenna gain, radiation pattern, return loss, and size miniaturization. Limited by the intrinsic physiochemical properties of conventional materials, however, it is still difficult to achieve satisfactory performances for antennas featuring flexibility and durability.

Most of the studies relating to wearable antennas are based on metal materials such as copper, silver, and aluminum [24,25,26]. These materials, however, suffer from unsatisfactory flexibility, which hinders their applications in wearable devices [27,28,29]. Textile-based conductive materials have been explored for realizing wearable antennas. Although these materials render better flexibility for the antenna, they cannot function properly in humid and high-temperature environments [30,31]. Recently, conductive polymers and traditional carbon-based materials have been explored as conductive materials instead of metals to fabricate conformal antennas [32,33,34,35]. These materials have good flexibility and are chemically inert, but suffer from mediocre conductivity, which makes it difficult for the antennas to achieve good radiation characteristics in the RF region [36,37]. At the same time, the film-forming process of these materials is complicated and this leads to prohibitive cost.

Due to the above-mentioned limitations of the common materials used for antenna fabrication, most of the proposed wearable antennas exhibit a high profile and poor flexibility, which make it difficult for the antennas to be integrated with on-body devices and to conform to the human body. Among existing UWB antenna research, Yimdjo et al. described an all-textile UWB antenna with a full-ground plane but it was limited for off-body applications due to the existence of the 2 mm-thick substrate [38]. A tapered monopole antenna with microstrip feed was proposed by Locher et al., using copper as the radiator material [39]. Suffering from the poor flexibility of copper, the antenna performance deteriorated as the antenna continued to bend, which makes it infeasible to be applied in wearable devices. On the other hand, a highly conductive graphene-assembled film (GAF) was previously reported by our group with superb flexibility, light weight, and most importantly, a conductivity of ~10^6^ S/m, which is comparable to that of the traditional metal materials. The GAF has been demonstrated to be a good alternative material for wearable antennas [40,41]. Table 1 provides a comparison between the proposed antenna and antennas described in some published works.

In this article, a flexible UWB GAF antenna fed by a CPW structure for wearable applications is presented and investigated. The antenna is easy to fabricate with a one-step laser-direct molding engraving method. As the thickness of the substrate dielectric material is only 255 μm and a CPW structure without background is used, an antenna with a very low profile of only 0.28 mm can be achieved. The measured bandwidth of 67% (4.0–8.0 GHz) is achieved by the two H-shaped slots etched from the ground. The proposed antenna shows a bending insensitive UWB characteristic, which guarantees its performance when applied in different scenarios.

The arrangement of the article is as follows. Section 1 introduces the requirements of wearable antennas, UWB technology, and the research status of the materials used to make wearable antennas. Section 2 contains detailed descriptions of material synthesis and characterization. Section 3 describes the design principle of the UWB antenna and also simulated and measured results of the proposed GAF UWB antenna. The performances of the GAF UWB antenna under different wearable application scenarios are presented in Section 4, which is followed by the conclusions.

## 2. Preparation and Characterization of GAF

The GAF was fabricated by the following steps [44]. Firstly, commercially available 50% wt graphene oxide (GO) slurry was diluted with ultrapure water to a final concentration of 20 mg/mL. The diluted GO suspension was then uniformly coated onto a polyethylene terephthalate (PET) substrate and evaporated for 24 h at room temperature to obtain the GO assembled films (GOAFs). Secondly, the GOAFs were annealed at 1300 °C for 120 min and then 2850 °C for 60 min in an argon (Ar) gas flow environment. Finally, the flexible GAFs were obtained by a rolling compression process. The prepared GAF can be folded into a crane without cracking, as shown in Figure 1a, demonstrating superb flexibility. The cross-sectional SEM image shows a final thickness of ~21 μm (Figure 1b). The microfold structures on the GAF surface shown in Figure 1c protect the GAF from cracking under severe deformations. Figure 1d shows the X-ray diffraction (XRD) pattern of the GAF with the characteristic (002) peak located at 2θ = 26.5°, indicating an interlayer spacing of ~0.336 nm. The diffraction peak (004) indicates a high degree of graphitization in the GAF material, which is further confirmed by the Raman spectrum (inset of Figure 1d). The sharp G peak (1585 cm^−1^) suggests a high level of sp^2^ hybridized carbon atoms, with the small D peak (1335 cm^−1^) and the strong G’ peak (2703 cm^–1^) demonstrating less lattice defects. We also tested the stability of the GAF by bending it repetitively 300 times while measuring the relative resistance through a digital multimeter (Agilent U1242B). The results are plotted in Figure 1e, showing a constant relative resistance during the test. In addition, a photograph and cross-sectional SEM image of the ceramic substrate are also shown in Figure 1f, demonstrating a thickness of ~255 μm and good flexibility, which is crucial for realizing wearability.

## 3. GAF Antenna Design and Measurement

The proposed GAF antenna is composed of a rectangular patch and two rectangular grounds as shown in Figure 2a. The H-slots etched on the ground are introduced to change the current distribution, hence achieving ultra-wide bandwidth [45,46]. For a CPW antenna, the required width of the slit width g and strip width w can be derived from the following formula [47]:(1)$$\varepsilon_{re}=1+\frac{\varepsilon_{r}-1}{2}\frac{K\left(k_{2}\right)}{K’\left(k_{2}\right)}\frac{K’\left(k_{1}\right)}{K\left(k_{1}\right)}$$
(2)$$Z_{0}=\frac{30\pi }{\sqrt{\varepsilon_{re}}}\frac{K’\left(k_{1}\right)}{K\left(k_{1}\right)}$$
(3)$$\frac{K\left(k\right)}{K’\left(k\right)}=\left\{\begin{array}{c} \frac{\pi }{\ln\left[2\left(1+\sqrt[{4}]{1-k^{2}}\right)/\left(1-\sqrt[{4}]{1-k^{2}}\right)\right]}\;0\leq k\leq 0.707\\ \frac{\ln\left[2\left(1+\sqrt{k}\right)/\left(1-\sqrt{k}\right)\right]}{\pi }\;0.707\leq k\leq 1 \end{array}\right.$$
(4)$$k_{1}=\frac{a}{b}$$
(5)$$k_{2}=\frac{\sinh\left(\pi a/2h\right)}{\sinh\left(\pi b/2h\right)}$$
where w and g are the strip width and slit width, respectively, with w=2a, g =b−a; ε_r_, ε_re_, and *h* are the relative dielectric constant, effective dielectric constant, and dielectric plate thickness, respectively; *K*(*k*) and *K′*(*k*) are the first type ellipse integral function and its supplementary function; *k* is an independent variable. Based on the design theory of planar monopole antennas, a rectangular radiation patch with resonance frequency *f* is designed, in which the patch length is *L*_1_ and the width is *W*_1_.[48]
(6)$$L_{1}=\frac{c}{2f\sqrt{\frac{\varepsilon_{r}+1}{2}}}$$
(7)$$W_{1}=\frac{c\sqrt{2}}{2f\sqrt{\frac{\varepsilon_{r}+1}{2}}}$$

The required width of the microstrip line is 1.9 mm for our substrate with a thickness of 255 μm and dielectric constant of 3.2, which is challenging to most of the commonly used fabrication techniques. In this paper, the problem of narrow microstrip line width is circumvented by introducing the CPW coplanar-waveguide feeding, whose width can be calculated with the above formulas (1–5). Importantly, with the proposed design, antennas with low profile can be achieved. The width *W_0_*, the gap *g*, and the dimensions of the patch (*W_1_* × *L_1_*) are optimized to achieve a 50 Ω characteristic impedance, which results in a final dimension of 32 × 52 × 0.28 mm^3^. The parameters of the H-slot (*L_3_*, *L_4_*, *L_5_*, and *L_6_*) are also optimized by applying electromagnetic simulations to ensure proper resonant frequency of the antenna. The optimized parameters are shown in Table 2.

The simulated |S_11_| using CST STUDIO SUITE of the optimized antenna is plotted in Figure 2b. It can be seen that the |S_11_| of the antenna has a −10 dB bandwidth from 4.3 GHz to 8.0 GHz, indicating a relative bandwidth of 60% with a maximum gain of 4.7 dBi. Figure 2c shows the simulated far-field radiation patterns of the antenna at 5.6 GHz, demonstrating a low backward radiation. In order to explore the performance of the antenna under bending conditions, electromagnetic simulations were carried out to determine the performance change of the antenna under various bending angles (θ) with the simulation model shown in Figure 2d. The overall spectrum shape of the simulated |S_11_| curves under different bending angles of θ = 0°, 30°, 60°, and 75° remains unchanged as depicted in Figure 2e, which indicates that the performance of the antenna is pretty robust against the effect of bending. It is worth noting that as the antenna bends, the shapes of the radiator and the ground change accordingly. This results in a different radiation pattern as we can see in Figure 2f, where we achieve better directionality with the antenna bending at θ = 60°. In order to investigate the effects of the H-shaped slot in realizing ultra-wide bandwidth, the vector plots of current distribution of the suggested antenna at three different frequencies are displayed in Figure 2g. The surface current mainly distributes at the patch edges and the H-slot of the ground plane along the y-axis. This indicates that the portion of the ground plane that is close to the patch can be considered as a part of the radiating structure. Therefore, adding slots enhances the radiation performance. As can be seen in Figure 2g, the surface current density is noticeably increased at the feed line at 7.1 GHz, indicating the significant effect of the feed line at high frequency. Figure 2h is the simulation result of the radiation efficiency of the antenna. It can be seen that as the bending angle increases, the radiation efficiency gradually decreases.

The GAF antenna prototype was fabricated by a one-step laser-direct molding engraving method with high resolution (Figure 3a) and fed through a standard 50 Ω SMA connector. The results were measured using Network Analyzer (PNA, Keysight N5247A). The measured |S_11_| is given in Figure 3b (red line) with the simulation result (blue dashed line) included as a comparison. It can be seen that the measured results are in good agreement with the simulation results. The actual bandwidth of the antenna covers 4.1–8.0 GHz, with the best impedance matching achieved at 4.45, 5.6, and 7.1 GHz. The resonant frequency of the GAF antenna is slightly off compared to the simulation result, which may be caused by the parasitic effect. Due to the influence of parasitic elements, a parasitic inductance, capacitance or resistance may be generated in a place where the inductance, capacitance or resistance was not originally designed. During the processing of the antenna, the silver glue used for soldering may produce parasitic effects. In measurement, because the antenna needs to be connected to the SMA connector, it can induce some errors in the actual test results. In addition, because the antenna is omnidirectional, the network coaxial cable may also, to some extent, affect the antenna. This little discrepancy, however, will not affect the proper functioning of the antenna because the |S_11_| is still under −10 dB within the entire frequency range.

In practical wearable applications, the antennas should remain electrically stable under bending conditions. Therefore, the effect of antenna bending was evaluated. The resulting |S_11_| curves are depicted in Figure 3c. It can be observed that both the lower bandwidth and the upper bandwidth broaden as the antenna bends, whereas the radiation efficiency is not noticeably affected by bending with insignificant changes observed for different bending angles ranging from 0° to 75°. The UWB characteristic of the antenna is also not sensitive to bending as the |S_11_| values remain below −10 dB within 4.1–8.0 GHz under different bending angles.

Far-field radiation performances of the antenna were also measured for flat and bent situations. These experiments were conducted in an anechoic chamber, which is shown in Figure 3d. It can be seen from Figure 3e that the beam focuses more on the target directions when the antenna is in its bent state with the back lobe getting narrower and featuring a higher front-to-back ratio. Figure 3f demonstrates that the gain increases with a bending angle of θ = 60° against the flat state across the entire measured frequency range, which is advantageous for on-body communications.

## 4. Measurement of GAF Antenna in Wearable Applications

For a wearable antenna, the effect of the human body and clothes should also be considered since these devices often need to work in close proximity to a human body. To test the feasibility of the proposed GAF antenna for wearable devices and smart clothing, we measure the reflection coefficients (|S_11_|) of the fabricated antenna under different application scenarios as shown in Figure 4a–c. The |S_11_| responses corresponding to the situations in Figure 4a–c are depicted in Figure 4d. When the antenna is conformed to the object’s wrist and arm in the bent condition, the impedance matching of the antenna is affected. The upper frequency resonance point of the antenna when on the wrist shifts to 7.5 GHz, and the mid-frequency resonance point of the antenna when attached to the back of the hand shifts to 5.2 GHz. When the antenna is integrated with clothes, the −10dB bandwidth of the antenna widens and the resonance point shifts left. Though the resonant frequencies of the antenna shift dramatically under different scenarios, the UWB characteristic of the proposed antenna ensures consistent performances with |S_11_| values lying below −10 dB within the operational range for on-body applications.

## 5. Conclusions

A CPW-fed flexible wearable antenna based on a highly conductive GAF and super-flexible composite ceramic material has been proposed. The H-shaped slots increase the bandwidth of the antenna so that the antenna can work properly within the ultra-wide range from 4.0 to 8.0 GHz with a maximum measured gain of 4.1 dBi. The CPW-fed structure effectively reduces the height of the antenna and thus increases the conformal ability of the antenna to the human body as well as the integratability with clothes. The designed antenna is compact in size and flexible, and is demonstrated to be pretty robust. It features an ultra-wide bandwidth characteristic that is insensitive to bending deformations and the change of surrounding media, which makes it a good candidate for wearable applications.

## Figures and Tables

**Figure 1 sensors-20-02552-f001:**
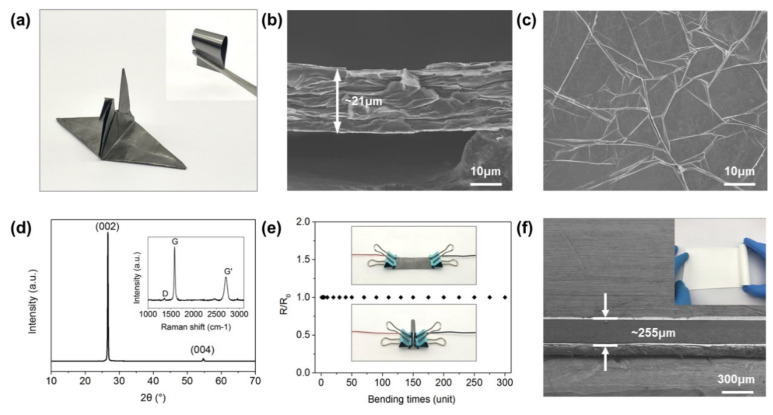
(**a**) Digital photographs of the graphene-assembled film (GAF) crane. (**b**) Cross-sectional and (**c**) top-view SEM images of GAFs. (**d**) XRD pattern and Raman spectrum (inset) of the GAF. (**e**) Mechanical stability test results of the GAF. (**f**) Cross-sectional SEM image and digital photograph (inset) of the substrate.

**Figure 2 sensors-20-02552-f002:**
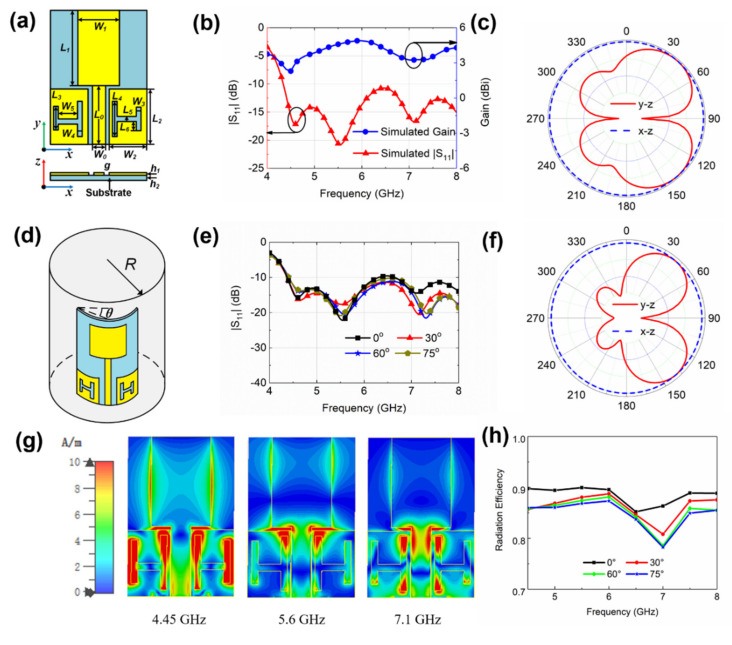
(**a**) Schematic of the GAF antenna. (**b**) Simulation results of return loss (red) and gain (blue) of the GAF antenna. (**c**) The simulated far-field pattern of the antenna at 5.6 GHz. (**d**) Simulation sketch of bending of the antenna in electromagnetic simulation software. (**e**) Simulation results of |S_11_|with different bending angles. (**f**) The simulated far-field patterns of the antenna with a bending angle of 60° at 5.6 GHz. (**g**) Current distribution of the antenna at 4.45, 5.6, and 7.1 GHz. (**h**) Simulation results of radiation efficiency with different bending angles.

**Figure 3 sensors-20-02552-f003:**
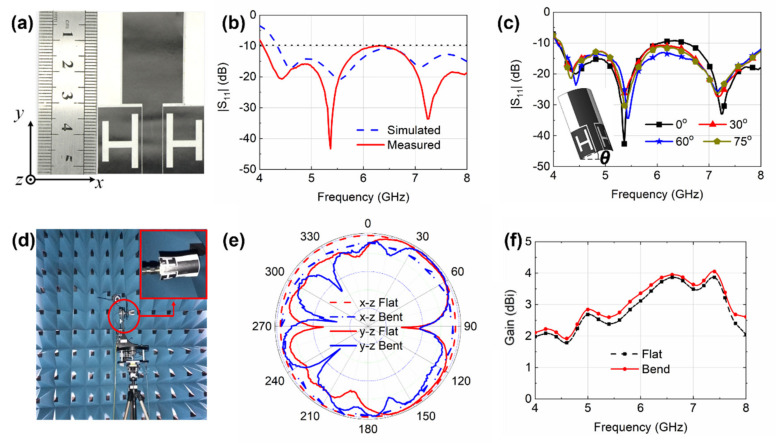
(**a**) The fabricated GAF antenna prototype. (**b**) Simulated (dashed blue line) and measured (red line) |S_11_| curves of the GAF antenna. (**c**) The measured |S_11_| curves of the GAF antenna with various bending angles of 0°, 30°, 60°, and 75°. (**d**) The anechoic chamber for radiation pattern measurement. (**e**) The measured radiation patterns of the antenna at 5.6 GHz in its flat (red) and bent (blue) state. (**f**) Measured gain of the antenna in its flat (black) and bent (red) state.

**Figure 4 sensors-20-02552-f004:**
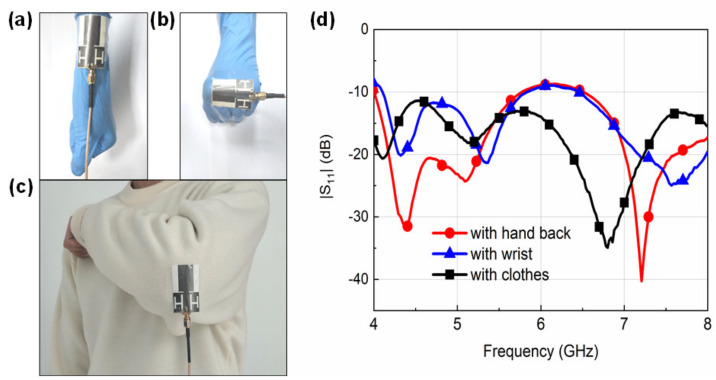
(**a**–**c**) Antennas under different application scenarios: (**a**) attached to the wrist, (**b**) attached to the back of the hand, (**c**) clipped onto clothes. (**d**) |S_11_| curves of the GAF antenna when attached to the back of the hand (red), wrist (blue), and clothes (black).

**Table 1 sensors-20-02552-t001:** Comparison of the proposed antenna with some references.

Ref.	Size (mm)	Frequency (GHz)	Conductive Material (S/m)	Substrate Material (ɛ_r_)	𝜂 (%)
[42]	80 × 67 × 3.4	3.7–10.3	nickel–copper–silver-coated nylon (1.02 × 10^5^)	PDMS (2.7)	27
[38]	80 × 61 × 4.51	2–12	ShieldIt conductive textile (1.18 × 10^5^)	Felt (1.45)	N/A
[43]	13 × 38 × 1.8	3–12	Copper (10^7^)	Polyimide (3.5)	N/A
[15]	30 × 40 × 1.15	2.8–16	Copper (10^7^)	Cotton (1.75)	50
This work	32 × 52 × 0.28	4–8	GAF (10^6^)	Ceramic (3.2)	90

**Table 2 sensors-20-02552-t002:** The optimized parameters of the antenna.

Parameters	Value (mm)	Parameters	Value (mm)
*W_0_*	6	*W_3_*	2
*L_0_*	23	*L_3_*	8
*W_1_*	18	*W_4_*	2
*L_1_*	29	*L_4_*	15
*W_2_*	13	*W_5_*	7
*L_2_*	20	*L_5_*	2
*h_1_*	0.025	*h_2_*	0.255

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
