# Peer review of "Compact and Low-Profile UWB Antenna Based on Graphene-Assembled Films for Wearable Applications"

_sensors, 2020, doi:10.3390/s20092552_

Round 1

Reviewer 1 Report

The manuscript "Compact and Low-Profile UWB Antenna Based on Graphene-Assembled Films for Wearable Applications" suitable for publication in sensors and can be accepted after correcting a few grammatical errors.

Author Response

Comment: The manuscript "Compact and Low-Profile UWB Antenna Based on Graphene-Assembled Films for Wearable Applications" suitable for publication in sensors and can be accepted after correcting a few grammatical errors.

Response: We thank the reviewer for the overall positive evaluation of our work. We do regret the typos and grammatical errors we made and we have corrected them in the revised manuscript.

Reviewer 2 Report

thank you for working on graphene-based UWB antenna for wearable applications. some of my major comments are as follows:

(1) Replace the term Letter all over manuscript like "In this Letter" in the abstract.

(2) Before explaining wearable a comprehensive paragraph need to be explained and introduced in regard of UWB antennas and their characteristics, Like (a) Compact CPW Fed Switchable UWB Antenna as an Antenna Filter
at Narrow-Frequency Bands (b) A study of UWB FM-CW radar for the detection of human beings in motion inside a building (c) Sensors 201818(10), 3330, and so on.

(3) Some latest wearable UWB antennas must be discussed in the introduction in detail and their drawbacks.

(4) The manuscript arrangement paragraph is missing and needs to be introduced.

(5) Line 60: Preparation spelling should be corrected.

(6) Equations(1) to (5) look like that the parameters cannot be extracted manually from it. Please show it how a particular antenna parameter is extracted from it. (show it in a table)

(6) Why radiation pattern is selected at 5.6 GHZ. for UWB three freq bands are must like lower, Mid, and upper freq bands.

(7) Show results at different bending angles by mentioning in tables and plots. Also, show the deviations in each case.

(8) Provide percentage radiation efficiency and realized antenna gain (dBi) in each case.

(9) Current distribution plots is also required.

(10) Finally, there enough typos and grammatical mistakes and need to be revised very carefully for it.

Reviewer 3 Report

The paper proposes a Graphene based substrate for an UWB antenna assembled into a ceramic substrate.  The document is well written using clear English with minor English misspellings (see final comments). Moreover, the paper has a good structure making it very easy to read.

 The authors may however introduce a number of improvements:

In Section 2 the antenna fabrication process may be referenced by identifying bibliographic references where similar procedures have been described. Even if not used for the purpose of fabricating an antenna.    

In Section 3 equations 1 – 5 are not described. Even if reference [37] describes the meaning of all variable the same should be made in this paper. Describing the meaning of all variables is important to improve paper readability.

Also in Section 3 the Antenna behaviour under different temperature and humidity conditions may also improve the paper as authors reference these two parameters in the introductory section of their letter.

Misplaced dot in line 47.  

Line 58 “…a bending…” should be used instead of “…an bending…”  

Author Response

The paper proposes a Graphene based substrate for an UWB antenna assembled into a ceramic substrate.  The document is well written using clear English with minor English misspellings (see final comments). Moreover, the paper has a good structure making it very easy to read.

 The authors may however introduce a number of improvements:

Comment 1: In Section 2 the antenna fabrication process may be referenced by identifying bibliographic references where similar procedures have been described. Even if not used for the purpose of fabricating an antenna.   

Response: We thank the reviewer for this valuable suggestion. The identifying reference where similar procedures have been described has been added to the revised Section 2: The GAF was fabricated by the following steps.[40]

The manufacturing method of GAF is similar to the previous report. [40]. Tang, D.; Wang, Q.; Wang, Z.; Liu, Q.; Zhang, B.; He, D.; Wu, Z.; Mu, S.; of Science, S. Highly sensitive wearable sensor based on a flexible multi-layer graphene film antenna. Sci. Bull. 2018, v.63, 54–59.

Comment 2: In Section 3 equations 1 – 5 are not described. Even if reference [37] describes the meaning of all variable the same should be made in this paper. Describing the meaning of all variables is important to improve paper readability.

Response: We are very grateful for the reviewer's valuable suggestion. In response to the reviewer’s suggestion, we added variable descriptions on page 3, line 98-100:

“…Where εr, εre, h, w, g are relative dielectric constant, effective dielectric constant, dielectric plate thickness, feeder width, and gap between feeder and ground; K(k), K′(k) Respectively the first type ellipse integral function and its supplementary function; k is the independent variable.…”

Comment 3: Also in Section 3 the Antenna behaviour under different temperature and humidity conditions may also improve the paper as authors reference these two parameters in the introductory section of their letter.

Response: Thank the reviewer for such a valuable suggestion, the influence of temperature and humidity on the antenna has been explored in our team in the previous work (Sci. Bull. 2018, 63, 54–59). The result shows that there is no significant change of the working frequency, which suggests a stable antenna sensor under environment with different temperatures. This is caused by the high thermal conductivity of 1,932.73 W/(m K) obtained in our previous work

Comment 4: Misplaced dot in line 47.

Response: We regret this mistake and we have made the correction in the revised manuscript.

Comment 5: Line 58 “…a bending…” should be used instead of “…an bending…” 

Response: We regret the mistake. We have corrected it in the revised manuscript.

Round 2

Reviewer 2 Report

although authors tried to address comments but after reading revised version I have again some concerns regarding the revised manuscript:

(1) The novelty of the proposed antenna needs to be highlighted in a much more detailed way by adding a comparison section with the recent state of the arts.

(2) "Measurement of GAF antenna in wearable applications" This section need to be much more detailed as it is the main section of the manuscript.

Author Response

Response to Reviewer

We thank the reviewer for going through our manuscript and providing valuable and thoughtful feedback, which we have addressed in the revised a manuscript. The below list summarizes our responses to the individual points raised by the reviewer.

Comment 1: The novelty of the proposed antenna needs to be highlighted in a much more detailed way by adding a comparison section with the recent state of the arts.

Response: In response to the reviewer’s suggestion, we added on page 2, line 55-68:

“… Due to the above-mentioned limitations of the common materials used for antenna fabrication, most of the proposed wearable antennas exhibit high profile and poor flexibility, which makes it difficult for the antennas to be integrated with on-body devices and to conform to human body. Among existing UWB antenna research, Yimdjo Poffelie et al. described an all-textile UWB antenna with full-ground plane but limited for off-body applications due to the existence of the 2 mm-thick substrate[38]. A tapered monopole antenna with microstrip feed is proposed by Locher, I et al. using copper as the radiator material[39]. Suffering from the poor flexibility of copper, the antenna performance deteriorates as the antenna continues to bend which makes it infeasible to be applied in wearable devices. On the other hand, a highly conductive graphene-assembled film (GAF) is previously reported by our group with superb flexibility, light weight and most importantly, a conductivity of ~106 S/m which is comparable to that of the traditional metal materials. The GAF is demonstrated to be a good alternative material for wearable antenna[40,41]. Table I provides a comparison between the proposed antenna and antennas described in some published works…”

TABLE I Comprison of the proposed antenna with some references

Ref.

Size

(mm)

Frequency (GHz)

Conductive material (S/m)

Substrate

material (ɛr)

?(%)

[42]

80×67×3.4

3.7-10.3

nickel–copper–silver-coated nylon (1.02×105)

PDMS (2.7)

27

[38]

80×61×4.51

2-12

ShieldIt conductive textile(1.18 ×105

Felt (1.45)

N.A

[43]

13×38×1.8

3-12

Copper (107)

Polyimide (3.5)

N.A

[15]

30×40×1.15

2.8-16

Copper (107)

Cotton (1.75)

50

This work

32×52×0.28

4-8

GAF (106)

Ceramic (3.2)

90

As suggested, the above new literatures have been added into the introduction and reference list:

  1. Yimdjo Poffelie, L.A.; Soh, P.J.; Yan, S.; Vandenbosch, A.E.G. A High-Fidelity All-Textile UWB Antenna with Low Back Radiation for Off-Body WBAN Applications. IEEE Trans. Antennas Propag. 2016,654,757-760.
  2. Locher, I.; Klemm, M.; Kirstein, T.; Trster, G. Design and characterization of purely textile patch antennas. IEEE Trans. Adv. Packag. 2006, 29, 777–788.
  3. Simorangkir, R.B.V.B.; Kiourti, A.; Esselle, K.P. UWB Wearable Antenna with a Full Ground Plane Based on PDMS-Embedded Conductive Fabric. In Proceedings of the IEEE Antennas and Wireless Propagation Letters; 2018; pp. 493–496.
  4. Xu, L.J.; Wang, H.; Chang, Y.; Bo, Y. A flexible UWB inverted-F antenna for wearable application. Microw. Opt. Technol. Lett. 2017, 59, 2514–2518

Comment 2: "Measurement of GAF antenna in wearable applications" This section need to be much more detailed as it is the main section of the manuscript.

Response: In response to the reviewer’s suggestion, we added on page 8, line 190-193:

“…When the antenna is conformed to the object’s wrist and arm in bent condition, the impedance matching of the antenna is affected. The upper frequency resonance point of the antenna when on the wrist shifts to 7.5 GHz, and the mid frequency resonance point of the antenna attached to the back of the hand shifts 5.2 GHz. When the antenna is integrated with clothes, the -10dB bandwidth of the antenna widens and the resonance point shifts left…”